# Utility of Kolmogorov complexity measures: Analysis of L2 groups and L1 backgrounds

**Alaa Alzahrani** [ID] *

King Saud University, Riyadh, Saudi Arabia

* alzahrani.alaaa@gmail.com

**Data Availability Statement:** The data used are publicly avaliable corpora. All data analysis R scripts are avaliable at https://osf.io/56fn2/.

**Funding:** The author(s) received no specific funding for this work.

## Abstract

The proliferation of automated syntactic complexity tools allowed the analysis of larger amounts of learner writing. However, existing tools tend to be language-specific or depend on segmenting learner production into native-based units of analysis. This study examined the utility of a language-general and unsupervised linguistic complexity metric: Kolmogorov complexity in discriminating between L2 proficiency levels within several languages (Czech, German, Italian, English) and across various L1 backgrounds ($N$ = 10) using two large CEFR-rater learner corpora. Kolmogorov complexity was measured at three levels: syntax, morphology, and overall linguistic complexity. Pairwise comparisons indicated that all Kolmogorov complexity measures discriminated among the proficiency levels within the L2s. L1-based variation in complexity was also observed. Distinct syntactic and morphological complexity patterns were found when L2 English writings were analyzed across versus within L1 backgrounds. These results indicate that Kolmogorov complexity could serve as a valuable metric in L2 writing research due to its cross-linguistic flexibility and holistic nature.

## Introduction

Complexity, along with Accuracy and Fluency (CAF), has been regarded as a key construct in Second Language (L2) writing research [1, 2], e.g., [3–7]. Empirical evidence suggests that complexity measures do not only index L2 writing quality, but also L2 writing development and L2 proficiency. A growing number of studies have revealed the ability of complexity measures to distinguish between low-rated and high-rated papers [8–15]. Other studies have shown that complexity measures can be reliable indices of L2 writing development over time e.g., [16, 17], c.f., [18–23]. Another line of research has observed the increased use of some complexity measures with increasing L2 proficiency, suggesting that complexity measures can be good indicators of L2 proficiency [24–29], e.g., [30–32].

The introduction of automated complexity tools has facilitated the study of complexity patterns across larger data sets e.g., [33]. Yet, these tools tend to be language-specific and mostly tailored for English e.g., [34–37]. The restricted research attention on a few languages could lead to the oversight of important insights or complementary perspectives from less-explored languages as there is some evidence that complexity develops differently across languages [38], e.g., [39]. A promising language-agnostic complexity measure from the information theory

**Competing interests:** The authors have declared that no competing interests exist.

literature: Kolmogorov complexity [40] may serve as a valuable metric and allow more research on a larger number of L2 groups. As such, this study examined the ability of Kolmogorov complexity to distinguish among L2 proficiency levels within different L2 groups and across L1 backgrounds to understand L2 speaker variation in L2 writing [41].

## Literature review

### Complexity in L2 writing

In the Second Language Acquisition (SLA) field, the term complexity has been used to indicate two distinct concepts: cognitive complexity and linguistic complexity [42–44]. In response, Bulté and Housen [42] have proposed a complexity framework to differentiate between two types of complexity: absolute and relative complexity. Both complexity forms can be used to describe the degree to which specific language features (e.g., lexical items, syntactic rules, phonological patterns) or their use can be challenging. While the relative approach defines language complexity in terms of language speakers, the absolute approach defines it in quantitative terms. Relative complexity refers to when language features pose challenges for language learners due to differences in learners' aptitude, memory capacity, motivation and L2 proficiency levels. Absolute complexity concerns the intrinsic formal or semantic-functional features of language (e.g., word form, word meaning, form-meaning mappings), independent from the learner, such as saliency, input frequency, redundancy and L1-L2 similarity. The examined Kolmogorov complexity metric falls within the second approach.

A traditional view of complexity in L2 writing is that it develops in a linear manner, suggesting incremental increases in complexity with increasing L2 proficiency. For example, based on previous research findings Norris and Ortega [2] proposed a three-stage development pattern of syntactic complexity in L2 writing in which learners are thought progress from clausal coordination to subordination and finally to clausal/phrasal elaboration. However, this linear view of development has been subsequently criticized as it is not consistent with the empirical observation that language development is likely a non-linear process e.g., [17, 45, 46]. This finding is in line with Larsen-Freeman's [41] dynamic systems framework which suggests that complexity in L2 writing might be susceptible to intra-individual variability and inter-individual variation, resulting in unique developmental patterns across and within L2 learners. In this view, non-native speakers of various L2s might show different complexity patterns in their L2 writings, influenced by the specific features of the target L2 or their L1 [47, 48], e.g., [49, 50]. This potential variability in complexity patterns in L2 writing highlights the need for examining complexity in a broader range of L2s and among L2 learners with different L1 backgrounds.

### Measures of L2 writing complexity

**Typical complexity measures.** A wide range of measures have been developed over the last two decades to examine syntactic complexity, with far fewer metrics for morphological complexity for a review, [51]. Earlier syntactic complexity indices are largely considered as global measures, while the later ones are viewed as fine-grained e.g., [10, 21, 27, 52]. Common global, length-based measures include length of production (e.g., mean length of T-unit), subordination (e.g., dependent clauses per T-unit), coordination (e.g., coordinate phrases per clause), sentence complexity (i.e., clauses per sentence), and particular structures (e.g., complex nominals per T-unit) e.g., [53]. Fine-grained syntactic complexity measures focus on phrasal elaboration such as noun pre-modifiers, attributive adjectives and prepositional post-modifiers e.g., [37, 54]. Meanwhile, there are fewer morphological complexity measures in the literature. Known measures include Morphological Complexity Index [55], Inflectional

Diversity [56], and Type/Family Ratio [57]. Both syntactic and morphological complexity measure types have been shown to discriminate between L2 learners at different proficiency levels [24, 55], e.g., [58].

Despite the observed advantage of current L2 complexity measures, the automatic tools to calculate the scores of these measures are largely language-specific. With the advance of learner corpora, there has been a shift towards using automated complexity tools e.g., [33]. These tools tend to be language-specific and cannot be directly applied to non-target languages. For instance, well-known automated complexity metrics such as Coh-Metrix [34], Syntactical Complexity Analyzer L2SCA; [36] and Tool for the Automatic Analysis of Syntactic Sophistication and Complexity TAASSC; [37] have been successfully used to study complexity in English texts, but they cannot be readily used to generate metrics for non-English texts.

In addition to the language-specificity issue of automatic complexity tools, there is some evidence that traditional units of measures in complexity research (e.g., clauses, sentences, t-units) can be difficult to delineate, especially when dealing with written learner language [59], e.g., [60]. A manual analysis of complexity measures in L2 Finnish writings suggested that learners' written production may deviate from those of native speakers, which makes it difficult to categorize their language into rigid production units [59]. This observed variability in learner language can defy clear segmentation into clauses, sentences, or T-units, influencing by this the reliability of some complexity measures particularly when computed manually. A similar influence of learner error on the reliability of automated complexity metrics has been found in a study by Châu and Bulté [60]. This study compared automated (L2SCA/TAASSC) and manual analyses of syntactic complexity in beginner and lower-intermediate level L2 English texts. Results indicated that learner errors significantly decreased the reliability of the automated measures. More flexible complexity measures are thus needed to account for the lack of cross-linguistic complexity measures and the reported variability in learner language.

**Kolmogorov complexity.**   In contrast to typical complexity metrics, information-theoretic-based complexity metrics are non-language-specific and relatively less subjective. In the information theory literature, complexity is typically quantified and measured in terms of description length and/or predictability using two measurements: Shannon entropy and Kolmogorov complexity [38, 50, 61, 62]. Relevant to this study is Kolmogorov complexity, which approximates how much information a specific text holds by measuring the length of the shortest possible description of this text [40]. To illustrate, consider example (1). Although both example strings have the same length (20 characters), only string (1a) is compact since it can be described as 10 times ab. Meanwhile, the length of the shortest possible description of string (1b) is the whole 20-character string. Thus, string (1b) is deemed more complex than string (1a) from the Kolmogorov complexity perspective. The Kolmogorov complexity of a text is determined by its unpredictability: the less predictable, the more complex.

1. a. abababababababababab
   b. ag!73kjrq4#tmn0e1y5z

Kolmogorov complexity is computationally difficult, so file compressors are used to approximate it [40]. Text compression software like gzip compresses text strings by encoding new strings based on previously encountered strings. As such, they measure the amount of information and redundancy in a specific text. In this study, Kolmogorov metrics will be used to examine complexity at three levels: morphological, syntactic, and overall linguistic complexity.

Kolmogorov complexity shares some similarities with existing complexity measures. First, both focus on the formal linguistic aspect rather than meaning or functions. Second, Kolmogorov complexity metrics are a holistic measure of complexity like typical length-based

complexity metrics (e.g., length of T-unit). Despite these overlaps, Kolmogorov complexity metrics remain unique in their ease of applicability as they can be readily used to gauge complexity across a wider number of typologically different languages [38]. Using a universal metric can help increase consistency in the study of L2 writing complexity while accommodating the observed distinct complexity trajectories in different languages e.g., [44]. Another advantage of Kolmogorov complexity metrics is their comparatively higher objectivity since they do not rely on segmenting learner language into predefined categories e.g., [50]. These unique characteristics might make Kolmogorov complexity a suitable measure to examine across and within L2 speaker variation in L2 writing. Despite their strengths, information-theoretic linguistic complexity measures have rarely been utilized in the L2 writing field [26], e.g., [50, 63].

## Effect of L2 proficiency on L2 writing complexity

A positive effect of L2 proficiency on L2 writing complexity has been reported in the literature. Most studies have reported that global-based syntactic complexity metrics could distinguish the different L2 proficiency levels in L2 written production [27, 53, 64, 65], e.g., [66]. However, a common criticism leveled against global/length-based metrics is that they may be better suited to capture complexity in spoken than written language [2, 42, 54, 67–69]. For example, Biber et al. [68, 69] suggested that spoken discourse typically exhibits clausal complexity, while academic writing generally features phrasal complexity.

Most studies investigating the predictive power of fine-grained syntactic complexity for L2 proficiency have focused on L2 English speakers [30, 31, 70–72], with few exceptions [32]. Although this body of work is informative, it is mainly based on writings from L2 English learners who might not show similar syntactic developmental patterns as L2 learners of non-English languages. One reason motivating research on L2 English learners is the availability of an English-based automated tool for analyzing fine-grained syntactic complexity TAASSC; [37].

While a sizable number of studies have explored the link between L2 proficiency and non-algorithmic syntactic complexity measures, far fewer studies have investigated this link using Kolmogorov complexity measures [26, 50]. A pioneering study by Ehret and Szmrecsanyi [50] examined the relationship between three Kolmogorov complexity metrics and the amount of L2 instruction using the International Corpus of Learner English (ICLE). Results showed that increased L2 instructional exposure predicted increased complexity at both the overall and morphological levels, but decreased syntactic complexity (i.e., more sentences with free word orders). Meanwhile, Wang et al. [26] analyzed 774 argumentative writings produced by Chinese-English bilinguals across three proficiency levels (A2, B1, B2) from the Common European Framework of Reference for Languages (CEFR) using Kolmogorov complexity metrics, traditional syntactic and morphological complexity metrics, and fine-grained syntactic complexity metrics. The study found that Kolmogorov overall and syntactic complexity could significantly discriminate between all adjacent L2 proficiency pairs, while the other metrics could not.

Meanwhile, few works have investigated the link between morphological complexity metrics and L2 proficiency e.g., [55, 73, 74]. Notably two Kolmogorov complexity studies showed that EFL learner writings showed increased morphological complexity at higher L2 proficiency levels [26, 50]. This suggests that more advanced L2 learners tend to use more inflectional/derivational forms compared to less proficient learners.

## Effect of L1 on L2 writing complexity

A few studies have explored variation in L2 writing complexity in L2 learners from different L1s, and most have addressed syntactic complexity without paying sufficient attention to

morphological complexity [47, 48], e.g., [49]. For instance, Lu and Ai [48] revealed that treating different L1s as a single L2 group may obscure L1-related differences in the syntactic complexity in L2 writing. By focusing on the L1 effect, 14 syntactic measures from Lu's SCA tool reliably discriminated among ICLE essays from seven L1 groups (Japanese, Chinese, Russian, French, Bulgarian, German and Tswana) as well as L1 English essays. This study suggests that examining L2 learners from different L1 backgrounds as separate groups may better reveal variations in syntactic complexity in L2 writing. However, existing studies have not examined the L1 effect on complexity at the morphological level.

Only one study so far examined the ability of Kolmogorov complexity metrics to distinguish between four different L1 backgrounds in the ICLE [50]. Results indicated that L1 background was a good predictor of overall Kolmogorov complexity, with German EFL learners composing more Kolmogorov-complex essays than French, Italian, or Spanish EFL learners. The separate analysis of the L1 German subset further revealed that morphological complexity numerically increased with increasing L2 instructional exposure, while syntactic complexity numerically decreased. Although Ehret and Szmrecsanyi's study [50] is one of the few studies investigating Kolmogorov complexity across L1 backgrounds, it should be noted that the used corpus was unbalanced across different combinations of L1s and L2 exposure levels. Another issue relates to using reported years of L2 instruction as an index of L2 proficiency. It is well-known that L2 learners may exhibit variability in their L2 proficiency despite experiencing comparable amounts of L2 language exposure e.g., [75]. As such, more investigation is needed to explore how Kolmogorov complexity measures can assess complexity across a broader range of L1 backgrounds in essays rated using a more objective L2 proficiency measure.

## The current study

Current automatic L2 complexity metrics can provide valuable information on writing performance and development. Yet, they are typically restricted to a subset of languages or depend on unreliable units of analysis to segment learner language. A potential measure that can be applied across various languages and requires minimal segmentation is Kolmogorov complexity—an information-theoretic, global complexity measure. The investigation of the utility of this measure is important as complexity patterns across language are not likely to be equal [38], e.g., [39], suggesting that findings shaped by L2 English learner data may not be readily generalizable to other L2 speakers. A cross-linguistic investigation of L2 writing complexity is thus desirable for a similar view, [76]. This comparatively language-agnostic measure might also serve as a universal complexity metric and improve consistency in measuring L2 complexity, a commonly reported issue in the literature e.g., [76]. However, limited attention has been paid to the use of Kolmogorov complexity to study complexity in the writings of different L2 groups and the potential influence of L1 background on L2 writing complexity. This study was set out to answer the following Research Questions (RQ):

1. Can Kolmogorov complexity metrics distinguish between L2 proficiency levels within distinct languages?

2. Can Kolmogorov complexity metrics distinguish between L2 proficiency levels across different L1s?

As it is argued that the syntactic complexity of L2 writings develops with increasing L2 proficiency [24], e.g., [25–27], this study investigated the reliability of Kolmogorov complexity by examining whether it can capture the expected syntactic differences across proficiency levels within various L2s (RQ1) and across L1 backgrounds (RQ2). The rationale for RQ1 is to establish the cross-linguistic flexibility of Kolmogorov complexity by examining whether it can

capture syntactic development across L2 proficiency stages in a wider set of languages rather than a single language. The motivation for RQ2 is to assess whether Kolmogorov complexity can detect L1-based differences in L2 writings similar to non-information theoretic complexity measures [47, 48], e.g., [49].

## Method

### Corpus

Two L2 written corpora were used in the present study: MERLIN [77] and the EF Cambridge Open Language Database (EFCAMDAT) [78–80]. MERLIN includes texts written by L2 Czech, German and Italian speakers, while EFCAMDAT covers texts written by L2 English speakers. These corpora were specifically chosen due to four reasons. First, they rate each text according to CEFR guidelines, which makes it possible to closely examine the relationship between Kolmogorov complexity and L2 proficiency level. Second, both corpora control the writing topic, allowing a more direct comparison of writing complexity within the same L2 group. Third, a sizable number of texts is found in each corpus, which could increase the reliability of statistical results [81, 82]. Finally, L2 speakers from different L1 backgrounds submitted their writings to EFCAMDAT, which could allow a robust investigation of the L1 effect.

The setting and writing genres in the two corpora were as follows. In MERLIN, L2 speakers had to complete a timed writing test, while in EFCAMDAT, writing was part of an untimed lesson activity. MERLIN mainly consisted of descriptive and expository writing tasks (e.g., writing letters/emails or describing a picture), whereas EFCAMDAT largely comprised persuasive texts (e.g., selling items in an online auction). These were the main genres in each corpus.

Written essays were selected based on two factors: L2 proficiency, and genre. For MERLIN, this study selected texts with A1 and B1 ratings because texts with higher or lower CEFR ratings were not equally represented across L2 in this corpus. As stated above, these texts fall in the descriptive and expository genres. For EFCAMDAT, as this corpus comprises many texts across four CEFR levels (A1, A2, B1, B2), they were all included. As mentioned above, EFCAMDAT essays were all persuasive essays. EFCAMDAT is semi-longitudinal and contains several texts written by the same L2 learner [78]. This study selected only one text per L2 learner to reduce idiosyncratic complexity features. While differences across corpora exist, this study focused on analyzing L2 writing complexity patterns within each L2 group. Descriptive summaries of selected texts from MERLIN are presented in Table 1, and from EFCAMDAT in Tables 2 and 3.

### Kolmogorov complexity metrics

Kolmogorov complexity is typically examined using three metrics: syntactic complexity, morphological complexity and overall complexity [26], e.g., [50]. These metrics are measured

**Table 1. Descriptive statistics of the Czech, German and Italian corpora by CEFR level.**

| | L2 | | | | | |
| --- | --- | --- | --- | --- | --- | --- |
| | Czech | | German | | Italian | |
| | A2 | B1 | A2 | B1 | A2 | B1 |
| No. of texts | 165 | 188 | 200 | 200 | 200 | 200 |
| Mean words (SD) | 93.51 (35.96) | 169.86 (54.12) | 58.09 (23.41) | 115.75 (47.23) | 67.46 (28.71) | 145.19 (47.77) |
| Mean sentences (SD) | 12.78 (6.34) | 16.39 (5.54) | 6.8 (3.28) | 11.26 (4.48) | 7.99 (3.28) | 12.02 (4) |
| Total words | 17579 | 28027 | 11619 | 23150 | 13492 | 29038 |

**Table 2. Descriptive statistics of the English corpus by CEFR level.**

|  | A1 | A2 | B1 | B2 |
|---|---|---|---|---|
| No. of texts | 200 | 200 | 200 | 200 |
| Mean words (SD) | 32.94 (13.22) | 58.49 (13.93) | 85.47 (20.85) | 113.68 (28.04) |
| Mean sentences (SD) | 5.10 (2.67) | 6.25 (2.61) | 7.24 (2.47) | 10.03 (4.16) |
| Total words | 6589 | 11698 | 17094 | 22736 |

**Table 3. Descriptive statistics of the English corpus by L1 and CEFR level.**

| L1 | CEFR level | | | | | | | |
|---|---|---|---|---|---|---|---|---|
|  | A1 | | A2 | | B1 | | B2 | |
|  | Mean words (SD) | Total words | Mean words (SD) | Total words | Mean words (SD) | Total words | Mean words (SD) | Total words |
| Arabic | 32(13) | 3490 | 52(15) | 5621 | 84(22) | 9119 | 102(29) | 11087 |
| French | 31(9) | 3407 | 58(13) | 6299 | 85 (16) | 9194 | 112(26) | 12111 |
| German | 33(13) | 3657 | 58(12) | 6348 | 89(17) | 9633 | 115(26) | 12485 |
| Italian | 35(16) | 3847 | 56(12) | 6068 | 83(15) | 8969 | 114(25) | 12363 |
| Japanese | 31(11) | 3417 | 55(10) | 6023 | 81(16) | 8781 | 109(25) | 11849 |
| Mandarin | 33(15) | 3593 | 59(16) | 6464 | 86(25) | 9375 | 113(28) | 12291 |
| Portuguese | 30(10) | 3264 | 58(13) | 6328 | 85(21) | 9221 | 109(30) | 11794 |
| Russian | 32(13) | 3457 | 60(12) | 6536 | 87(20) | 9459 | 119(26) | 12900 |
| Spanish | 30(10) | 3305 | 59(12) | 6374 | 80(19) | 8646 | 112(26) | 12200 |
| Turkish | 32(12) | 3456 | 53(13) | 5824 | 85(20) | 9241 | 106(26) | 11526 |

Note. There were 108 texts per language x proficiency level.

using an information-theoretic approach in which a text's complexity is assessed by the length of the shortest reproduction of it. This is typically done by compressing texts using data compression applications such as gzip. Texts with a smaller compressed size are considered less complex, whereas texts with a larger compressed size are likely to contain more complex linguistic features.

Calculation of Kolmogorov complexity was done in line with prior research [26, 50, 83]. Two main features distinguish the computation of Kolmogorov complexity: text distortion and text compression. First, texts are distorted for some Kolmogorov complexity metrics (syntactic and morphological complexity) but not for others (overall complexity). Text distortion tackles different linguistic elements depending on the type of complexity measure for a detailed discussion, see [83]. In syntactic complexity, 10% of all word tokens in each text are randomly deleted. In morphological complexity, 10% of characters in each text are randomly deleted. Second, texts must be compressed via software applications such as gzip to approximate the three Kolmogorov complexity metrics (syntactic, morphological, and overall complexity). To sum up, while syntactic and morphological complexity are calculated based on both text distortion and compression, overall complexity is calculated based on text compression.

Kolmogorov complexity was measured as follows. For syntactic and morphological complexity, the distorted and compressed file size in bytes is divided by the non-compressed file size in bytes. The interpretation of compression ratios differs between syntactic and morphological complexity [38, 50]. In terms of syntactic complexity, lower compression ratios after syntactic distortion indicate high Kolmogorov syntactic complexity. In the Kolmogorov complexity literature, syntactic complexity is associated with word order rigidity. Texts with rigid

word orders are thought to be more syntactically complex than texts with free word orders because distortion of fixed word order rules might introduce more noise. The increased random noise could compromise the compressibility of a text, resulting in lower compression for texts with fixed word order rules. On the other hand, a low compression ratio after morphological distortion means the text has low Kolmogorov morphological complexity because morphologically simple texts contain more non-inflected words, and they will be more affected by character deletion (distortion) than morphologically complex texts.

The overall complexity of texts is measured by conducting a linear regression analysis in which the independent variable is the uncompressed file size in bytes, and the dependent variable is the compressed file size in bytes. A regression analysis is computed to eliminate the correlation between the two file sizes [83]. Larger adjusted complexity scores (regression residuals) suggest higher overall Kolmogorov complexity. To increase statistical robustness, file distortion and compression was repeated 1000 times per L2 (e.g., all German texts). Random sampling was used so that a different part (character/word) of the text was distorted in each repetition.

## Kolmogorov complexity calculation tool

To our knwoledge, no specialized software currently automatically calculates Kolmogorov complexity metrics. This study calculated these metrics using R scripts developed and shared publicly by Ehret [83]. This calculation is done in several steps. First, the free compression program gzip must be downloaded and installed. Second, the corpus files should be stored in txt format. Third, Ehret's R scripts should be run, which would automatically calculate Kolmogorov complexity metrics. Further details about these R scripts can be found here: https://github.com/katehret/measuring-language-complexity.

## Data processing

The two corpora were cleaned in several steps following previous practice [26], e.g., [83]. First, all characters were converted to lowercase. Second, non-alphabetical characters (e.g., 1, $, #, +, ~) and punctuations (e.g., dashes, hyphens, commas) were removed. Third, all end-of-sentence markings (e.g., question marks, exclamation marks and semicolons) were replaced with full stops as they mark sentence boundaries in the calculation of Kolmogorov complexity. Finally, a manual check was performed to ensure the correct removal of non-alphabetical elements. These steps ensured better text compressibility and minimized the possibility of inflated complexity scores due to non-linguistic reasons.

## Statistical analyses

Significance testing was performed for two complexity metrics: syntactic and morphological complexity. Bayesian t-tests were used to examine differences in syntactic and morphological complexity metrics when there were two proficiency levels (A2, B1) in the corpus, while Kruskal-Wallis tests were computed when there were more than two proficiency levels (A1, A2, B1, B2). These two tests were used as they are relatively robust against violations of normality [84, 85]. T-tests were conducted using the R package BEST [86], while Kruskal-Wallis tests and their effect sizes were computed using the R packages stats [87] and rstatix [88]. For Bayesian t-tests, a result is considered significant when the estimate's CIs do not include zero [84]. As for the overall complexity metric, statistical tests would yield unreliable results for regression residuals. As such, only raw scores were reported. All analysis codes are available via the Open Science Framework at https://osf.io/56fn2/?view_only=438435909c914d668be06d50d5bff135.

**Table 4. Bayesian t-test results for syntactic and morphological Kolmogorov complexity by L2 and CEFR level.**

| Metric | L2 | | | | | |
|---|---|---|---|---|---|---|
| | Czech | | German | | Italian | |
| | Mean [95% CI] | Effect | Mean [95% CI] | Effect | Mean [95% CI] | Effect |
| Syn. | -.003 [-.004, -.003] | -.87 | -.005 [-.005, -.004] | -.84 | -.012 [-.012, -.011] | -1.93 |
| Morph. | .004 [.004, .005] | .80 | .012 [.011, .013] | 1.48 | .28 [.027, .030] | 2.48 |

Syn: syntactic complexity, Morph: morphological complexity, Effect size: Cohen's d.

## Results

The results of the first research question will be discussed first, followed by the results of the second research question.

### Kolmogorov complexity differences across L2s and L2 proficiency

**L2 Czech, German and Italian.** Table 4 summarizes results from Bayesian t-tests which revealed significant differences in all Kolmogorov complexity metrics between A2 and B1 levels per L2 language. As shown in Table 4, syntactic complexity significantly decreased from the beginner level (A2) to the intermediate proficiency stage (B1) in L2 Czech, German and Italian (Fig 1). Meanwhile, morphological complexity increased with increasing L2 proficiency (Fig 2). Overall, the analysis of L2 writings in Czech, German and Italian indicated that as L2

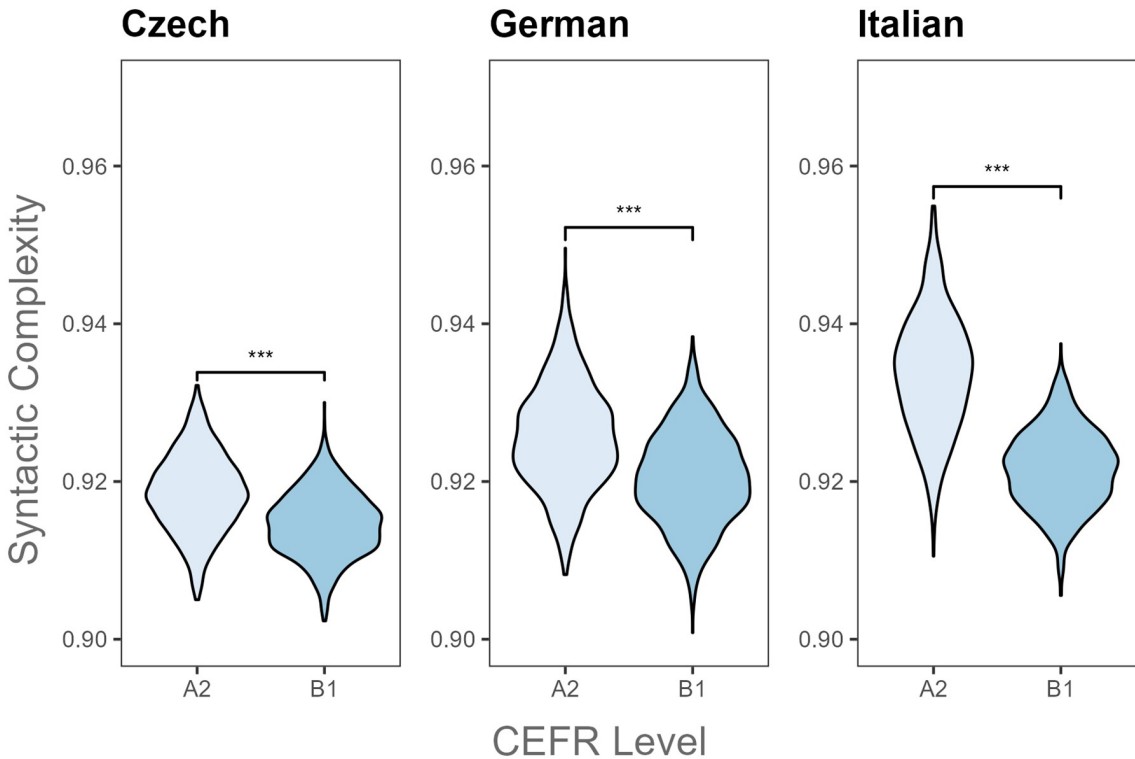

**Fig 1. Paired comparisons of syntactic complexity by CEFR level and L2 language.** *** $p \leq .001$.

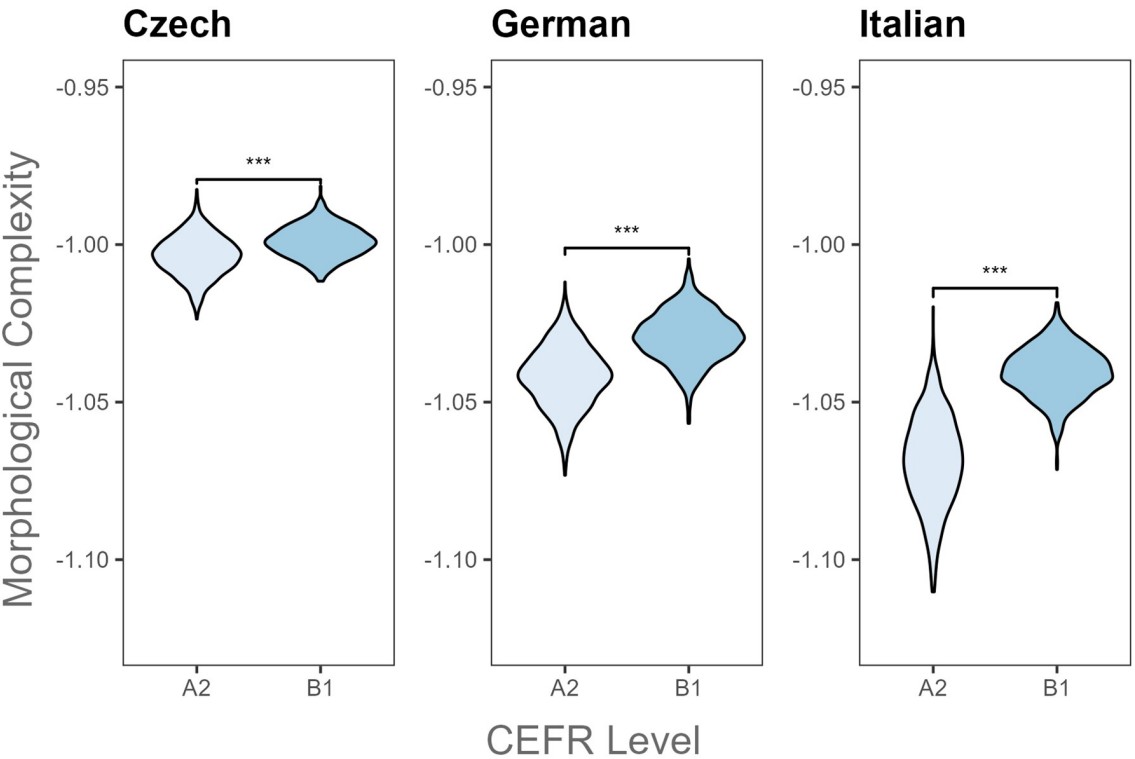

**Fig 2. Paired comparisons of morphological complexity by CEFR level and L2 language.** *** $p \leq .001$.

speakers progress in their L2, their writing becomes more complex in terms of morphology, but it becomes relatively syntactically simple.

Overall complexity metrics are summarized in Table 5. Increased L2 proficiency was associated with increased writing complexity in L2 German but not L2 Czech and Italian. L2 German speakers produced texts with higher overall complexity at the B1 level than A2. In contrast, L2 learners of Czech and Italian wrote less linguistically complex texts at B1 than at A2.

**L2 English.** In the L2 English corpus, there were significant differences in Kolmogorov syntactic and morphological complexity metrics across the four CEFR proficiency levels (Table 6). Syntactic complexity was statistically more pronounced in the upper-beginner and lower-intermediate English levels (A2, B1) compared to lower beginner and upper-intermediate levels (A1, B2) (Fig 3). In terms of morphological complexity, it was generally higher at the lower levels than higher ones. A1 level English learners produced more morphologically complex writing than A2 learners, with a similar difference between A2 and B1 CEFR levels (Fig 3). Yet, upper-intermediate L2 English writers (B2) tended to produce more morphologically complex texts than their lower-intermediate counterparts (B1). This increased use of

**Table 5. Overall complexity results by L2 and CEFR level ranked in a descending order.**

| L2 | CEFR | |
|---|---|---|
| | **A2** | **B1** |
| German | -3.56 | -2.72 |
| Czech | -2.30 | -3.09 |
| Italian | 0.07 | -2.94 |

**Table 6. Kruskal-Wallis results for Kolmogorov complexity metrics across four CEFR levels (A1, A2, B1, B2) in the 2 English corpus.**

| Metric | $\chi^2$ | Effect size |
|---|---|---|
| Syntactic | 401*** | .099 |
| Morphological | 2468*** | .616 |

Effect size = eta2[H].

*** $p \leq .001$.

morphologically complex word forms at A1 and B2 levels might indicate a U-shaped developmental pattern for morphological complexity among the examined L2 English learners.

Looking at overall complexity (Table 7), learners at the B2 CEFR level produce texts with the highest complexity score compared to the other examined levels. Surprisingly, A1-level English learners composed essays with higher overall complexity than those at B1 and A2 levels. Those at the A2 level produced the least linguistically complex texts. This very low complexity at the A2 stage might be traced back to the relatively low morphological complexity score for essays at that proficiency stage.

## Kolmogorov complexity differences across L1 background and L2 proficiency

The compression technique was applied 1000 times to each L1 background, with a random distortion of 10% of the sentence per text and iteration to examine the L1 effect. Similar to

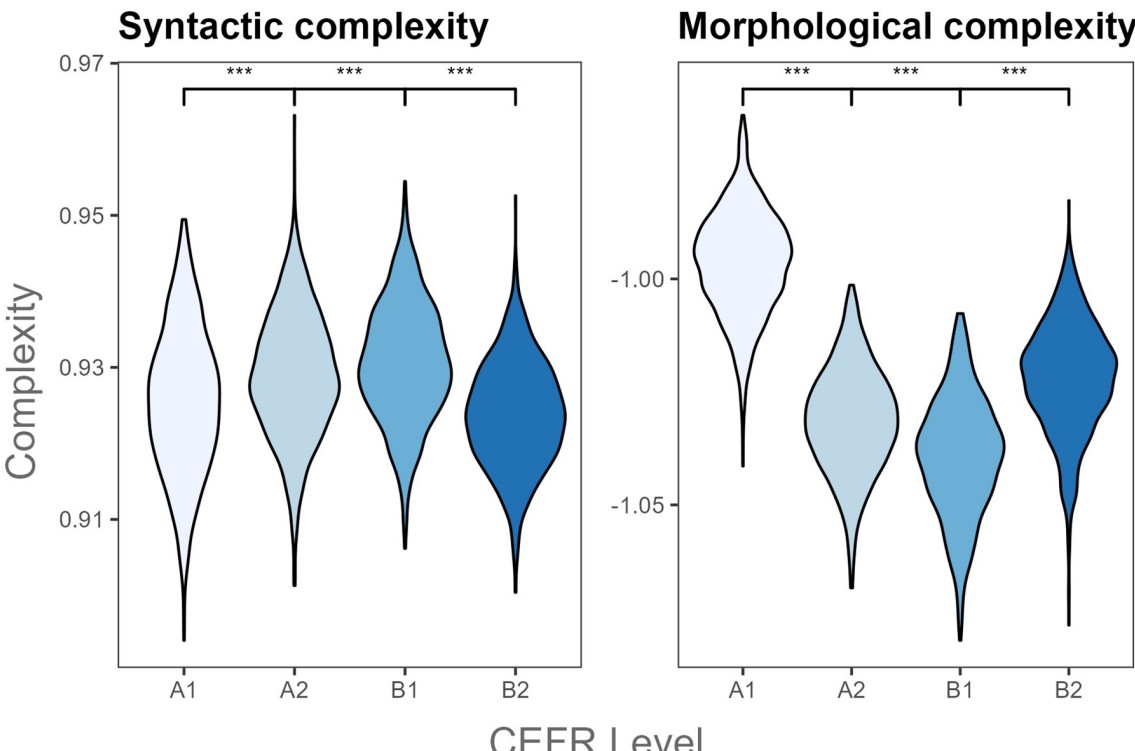

**Fig 3. Paired comparisons of syntactic and morphological complexity by CEFR level in L2 English writing.** *** $p \leq .001$.

**Table 7. Overall complexity results for L2 English texts by CEFR level ranked in a descending order.**

| CEFR | Adjusted overall complexity |
|------|------------------------------|
| B2 | 71.84 |
| A1 | 26.64 |
| B1 | -40.99 |
| A2 | -57.49 |

the L2 proficiency effect, this analysis compressed (1) undistorted, (2) morphologically distorted, and (3) syntactically distorted versions of the texts. The analysis showed that although some L2 groups showed overlapping complexity patterns in their L2 English writing, there were L1-based differences in each Kolmogorov complexity metric across L2 proficiency levels.

Table 8 presents results for syntactic and morphological complexity by L1 background. Starting with Kolmogorov syntactic complexity, all L1 groups showed differences in syntactic complexity between A1 and A2 proficiency levels. Some L1 groups showed no (L1 Turkish) or a slight difference (L1 Arabic), while most L1 groups experienced a change in the production of complex syntactic structures. Most L1 groups (French, Japanese, Mandarin, Portuguese, Russian, Spanish) preferred to write more syntactically complex texts at the A1 level compared to the A2 level. Nevertheless, other L1 groups such as German and Italian L1 speakers showed the opposite pattern: more syntactic complexity at A2 than at A1.

**Table 8. Kruskal-Wallis results for syntactic and morphological complexity across four CEFR levels (A1, A2, B1, B2) in the 2 English corpus by L1 background.**

| Complexity metric | L1 | $\chi^2$ | Effect size |
|-------------------|-----|----------|-------------|
| Syntactic | Arabic | 287*** | .071 |
| | French | 699*** | .174 |
| | German | 255*** | .063 |
| | Italian | 115*** | .028 |
| | Japanese | 625*** | .155 |
| | Mandarin | 492*** | .122 |
| | Portuguese | 156*** | .038 |
| | Russian | 794*** | .198 |
| | Spanish | 126*** | .030 |
| | Turkish | 9.53* | .001 |
| Morphological | Arabic | 168*** | .421 |
| | French | 214*** | .537 |
| | German | 174*** | .434 |
| | Italian | 185*** | .462 |
| | Japanese | 280*** | .701 |
| | Mandarin | 152*** | .380 |
| | Portuguese | 776*** | .193 |
| | Russian | 254*** | .637 |
| | Spanish | 739*** | .184 |
| | Turkish | 197*** | .493 |

* $p \leq .05$;

*** $p \leq .001$.

Similarly, there was variation in syntactic complexity between A2 and B1 CEFR levels across the L1 groups. Although most L1 groups showed differences in syntactic complexity between these two proficiency levels, Italian, Portuguese, and Turkish L1 speakers did not. A clear decrease in syntactic complexity is observed from level B1 to B2 texts across nine of the ten L1 groups. An exception is the Turkish L1 writers who tended to produce more syntactically complex essays at level B2 rather than at B1. Overall, all the ten L1 groups showed variation in syntactic complexity across the four proficiency levels (A1, A2, B1, B2), and some groups (Arabic, Italian, Portuguese, and Turkish L1 speakers) exhibited unique developmental syntactic complexity patterns (Fig 4).

A less prominent L1-effect is observed for the morphological complexity metric (Fig 5). Almost all L1 groups showed a U-shaped developmental use of complex morphological forms, with a tendency to produce more morphologically complex texts at A1 and B2 levels compared

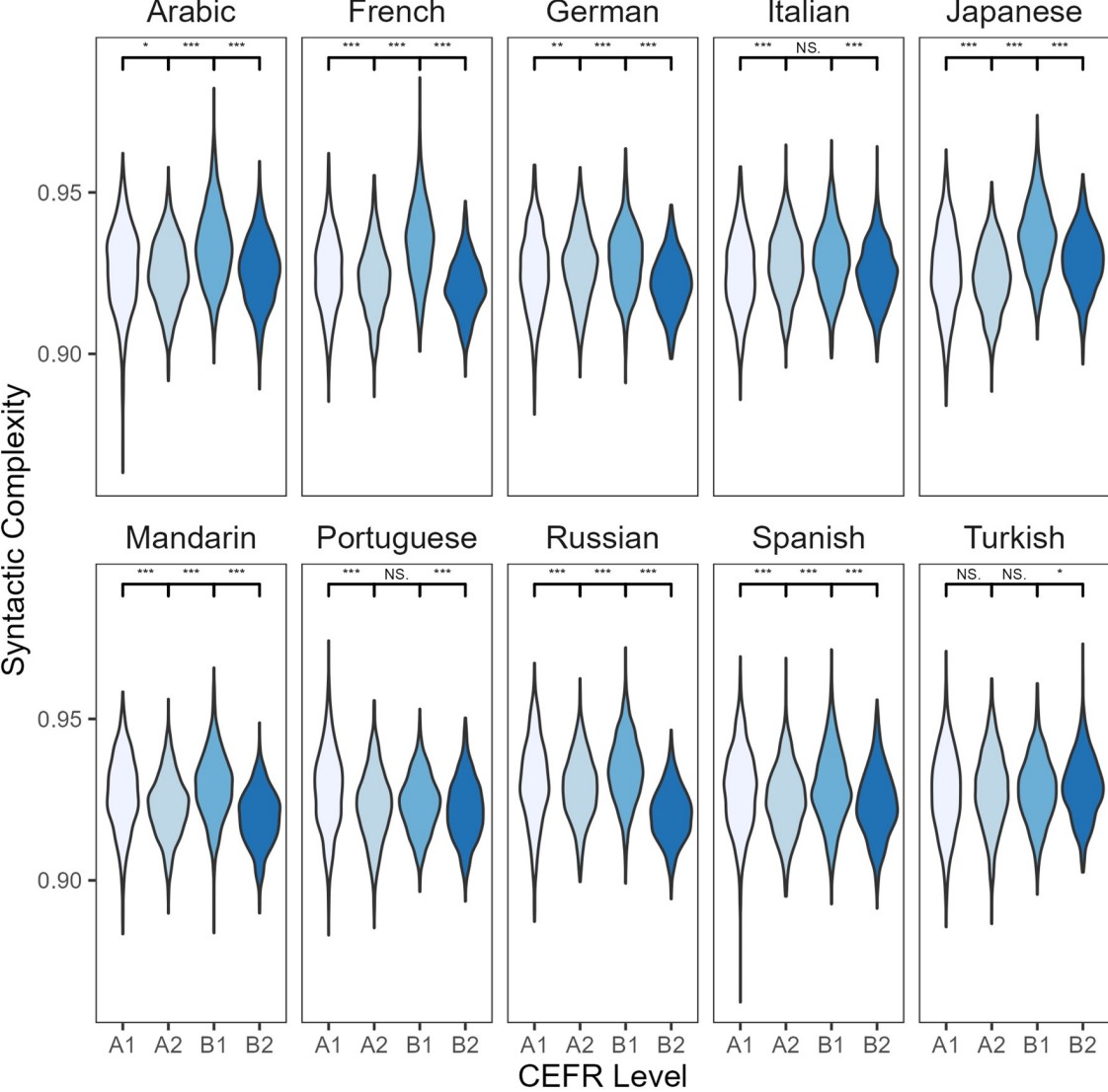

**Fig 4. Paired comparisons of syntactic complexity by L1 and CEFR level in L2 English writing.** $^*$ $p \le .05$; $^{**}$ $p < .001$; $^{***}$ $p \le .001$; NS, no significant difference.

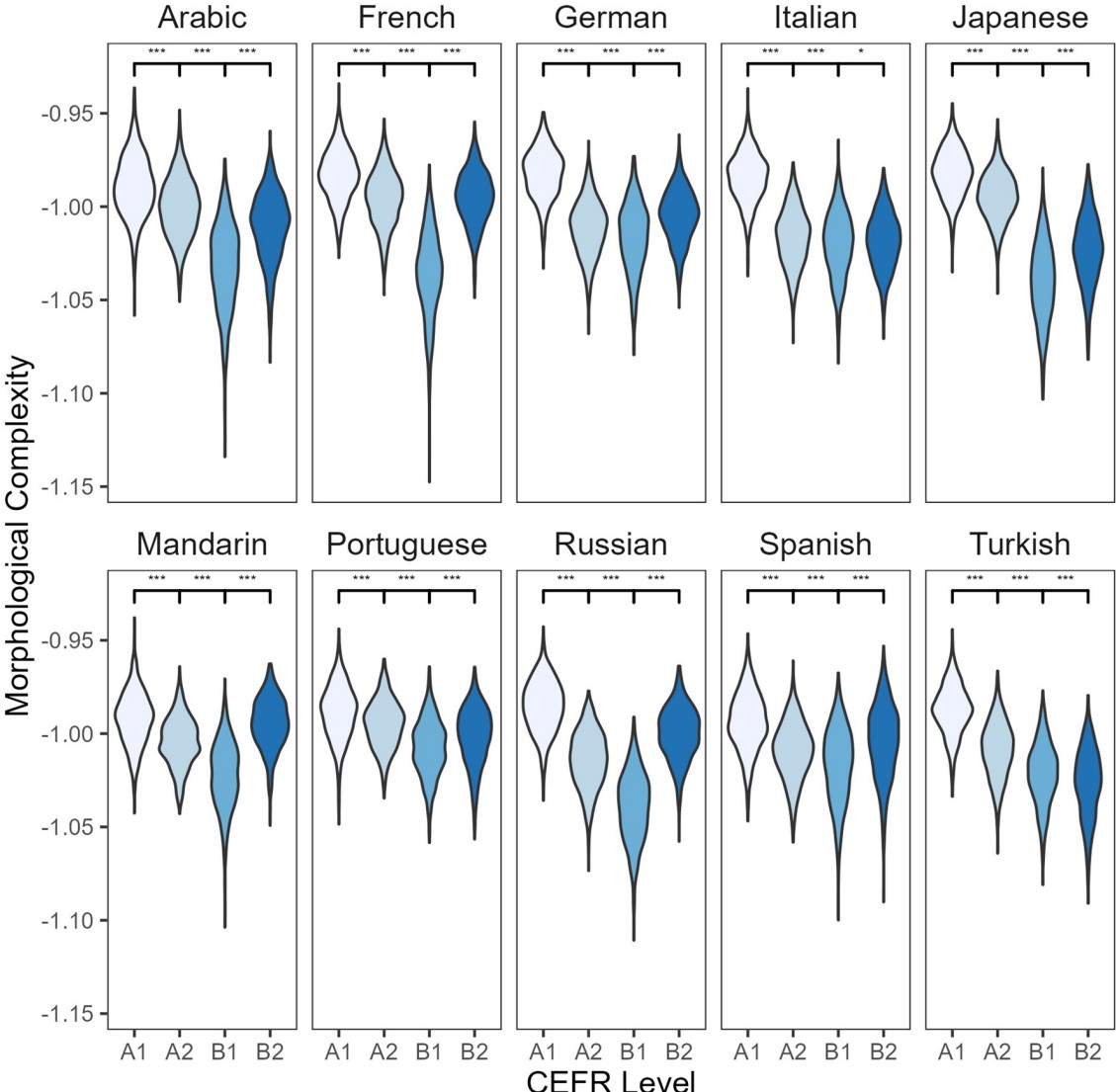

**Fig 5. Paired comparisons of morphological complexity by L1 and CEFR level in L2 English writing.** * $p \leq .05$; *** $p \leq .001$.

to A2 and B1, respectively. However, the L1 Turkish group diverged from this pattern. Turkish-English bilinguals continued to produce less morphologically complex sentences with increasing L2 proficiency. Further, the L1 Italian group demonstrated a small significant difference in morphological complexity between B1 and B2 levels compared to the other L1 groups.

L1-based variability was also found in the overall Kolmogorov complexity. Table 9 summarizes overall Kolmogorov complexity scores by L1 and proficiency level in English L2 writing data. The majority of English learners from different L1 backgrounds showed higher complexity at A1 than at A2 (German, Italian, Mandarin, Russian, Spanish, Turkish). However, some L1 groups such as L1 Arabic, French, Japanese, and Portuguese produced more complex writings at level A2 than A1 (Fig 6). As for the difference between A2 and B1 levels, all the L1 groups wrote more complex essays at the A2 proficiency stage than B1 except for L1 Italian and Turkish speakers. Finally, all groups produced more complex texts at B2 than B1 CEFR

**Table 9. Overall complexity results for L2 English texts by L1 background and CEFR level.**

| L1 | CEFR | | | |
| --- | --- | --- | --- | --- |
| | A1 | A2 | B1 | B2 |
| Arabic | -12.00 | 12.18 | -29.88 | 29.70 |
| French | 6.49 | 12.41 | -79.15 | 60.25 |
| German | 20.58 | -25.98 | -31.61 | 37.02 |
| Italian | 7.94 | -20.57 | -9.18 | 21.81 |
| Japanese | -14.54 | 39.54 | -52.16 | 27.16 |
| Mandarin | 4.12 | 0.05 | -59.59 | 55.42 |
| Portuguese | -6.07 | 5.27 | -17.04 | 17.84 |
| Russian | 24.76 | -16.34 | -96.74 | 88.32 |
| Spanish | -2.41 | -5.31 | -12.78 | 20.49 |
| Turkish | 4.86 | -7.48 | -0.40 | 3.02 |

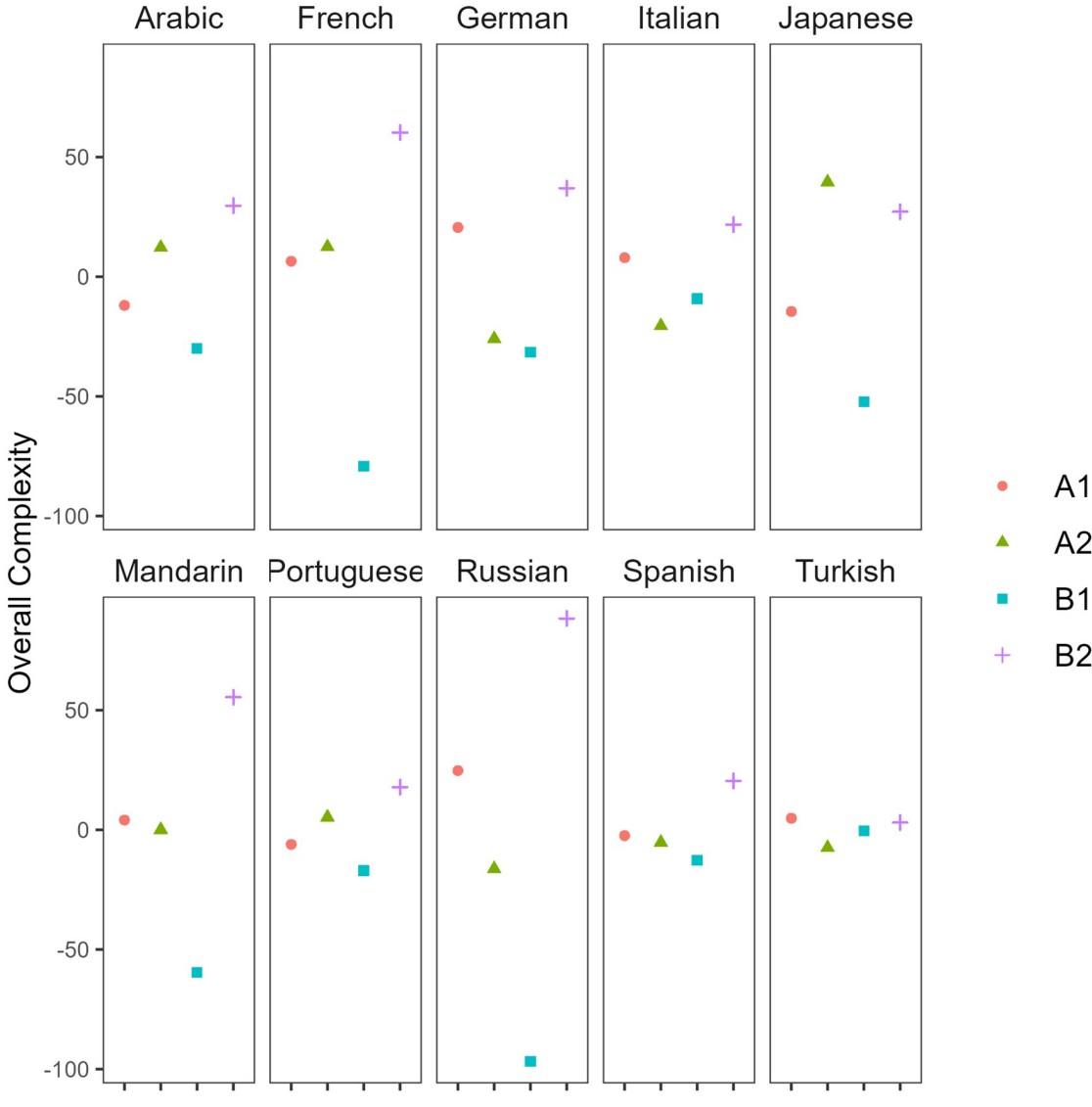

**Fig 6. Distribution of overall complexity scores by L1 and CEFR level in L2 English writing.**

level, suggesting limited L1-effects for overall Kolmogorov complexity at the B2 proficiency level.

## Discussion

Common computational tools for complexity metrics in L2 writing research are either tailored to specific languages e.g., [34–37] or may not reliably segment learner language especially at the lower proficiency levels [59], e.g., [60]. The language specificity problem has facilitated the study of writing complexity in predominant and resource-rich languages such as L2 English to the exclusion of other languages. The comparatively limited interest in other L2 speakers in the L2 writing complexity literature [76] may impede a comprehensive understanding of the relationship between writing complexity and L2 development across distinct languages. The reliability problem partly arises from the attempt to fit learner production into predefined units of analysis (e.g., clauses, T-units). A potential solution to these issues is using a language-general, unsupervised, information-theoretic measure of linguistic complexity: Kolmogorov complexity.

This study examined Kolmogorov complexity metrics' ability to distinguish between adjacent CEFR proficiency levels within L2s (RQ1) and across L1 backgrounds (RQ2). Three Kolmogorov complexity metrics (syntactic, morphological, and overall complexity) were examined in two large-scale and CEFR-labeled L2 corpora. Pairwise comparisons showed that Kolmogorov complexity discriminated among the proficiency levels differently within the L2s (Czech, German, Italian, English), suggesting its cross-linguistic flexibility. Likewise, there were differences in Kolmogorov complexity when analyzing L2 English writings across and within L1 backgrounds, indicating its utility in exploring L1-based differences in L2 writing.

### Kolmogorov complexity within L2

Unlike previous L2 Kolmogorov complexity research which mostly involved L2 English speakers [26, 50], the current study investigated how Kolmogorov complexity metrics could discriminate among L2 proficiency levels within four languages (Czech, German, Italian, English). Analyzing complexity in writings by L2 speakers other than English is important to ensure that findings from a larger pool of languages inform our understanding of the relationship between complexity and L2 proficiency. This investigation could also assess whether Kolmogorov complexity metrics can generate meaningful results for different languages. Three Kolmogorov complexity indices were investigated across two CEFR proficiency levels (A2, B1) in L2 Czech, German, Italian writings and four CEFR levels (A1, A2, B1, B2) in L2 English writings. If Kolmogorov metrics found differences between L2 proficiency levels within each examined L2, this would provide evidence for its cross-linguistic flexibility. On the other hand, if Kolmogorov metrics did not detect a significant difference between L2 proficiency levels for any of the examined L2, this would indicate that these measures are limited to specific languages.

Kolmogorov complexity measures were able to detect differences across L2 proficiency levels within the examined L2 groups. L2 Czech, German and Italian learners composed writings with similar syntactic and morphological complexity results. Beginner L2 Czech, German, and Italian learners produced more syntactically complex texts (i.e., more sentences with rigid word orders) but with limited morphological complexity compared to more proficient L2 learners. However, overall complexity results differed across these three languages. Texts written by less proficient L2 speakers of Czech and Italian had higher overall complexity than those written by more proficient L2 speakers, while the opposite result was found for L2 German speakers.

These findings suggest that lower-level L2 learners of Czech, German and Italian groups tended to produce sentences with fixed word orders that contained non-inflected words. However, with increasing proficiency, they composed texts with less rigid word-orders which included more inflected vocabulary items carrying more grammatical information such as tense, number and case features. The syntactic and morphological complexity results are in line with previous studies on L2 English writing, showing that increased L2 English proficiency was accompanied by increased Kolmogorov morphological complexity and reduced Kolmogorov syntactic complexity (i.e., more sentences with free word orders) in L2 written production [26, 50]. The current study extended previous English-based findings to three new L2 groups (Czech, German and Italian), suggesting that Kolmogorov complexity measures can capture variation in syntactic and morphological features in L2 learners' writings at different L2 proficiency levels. However, overall complexity results for L2 German writings replicated prior Kolmogorov complexity studies on English L2 writings [26, 50], while those for L2 Czech and Italian did not. Overall, these results indicate the effectiveness of Kolmogorov complexity metrics in highlighting differences in linguistic complexity across proficiency levels within each L2 group.

Additionally, Kolmogorov complexity metrics also differentiated between English L2 speakers from varying proficiency levels. Kolmogorov Syntactic complexity was highest in the upper-beginner and lower-intermediate English levels (A2, B1) than in the lower beginner and upper-intermediate levels (A1, B2), indicating an inverted U-shaped development. The opposite was found for Kolmogorov morphological complexity: a U-shaped pattern in the use of inflected words, with initial increased use at A1, followed by simplification at A2 and B1, and potential return to complexity at B2. Finally, Kolmogorov overall complexity indicated that texts produced by B2 English learners exhibited more overall complexity than those at the lower CEFR levels. Only the overall complexity findings were in line with prior research [26, 50], whereas the non-linear developments in syntactic and morphological complexity diverged from this body of research.

The divergence of L2 English results from earlier similar studies can be attributed to several factors. First, the examined corpora differed across the studies. For example, Wang et al. [26] focused on timed argumentative essays, while the present study included untimed persuasive essays. However, variation in the characteristics of corpora across studies might not be have contributed to the observed differences in findings as Ehret and Szmrecsanyi's [50] study analyzed untimed argumentative essays, yet their findings did not match the present study. A second potential explanation for the inconsistency in L2 English results can relate to the examined L2 proficiency levels. Wang et al. [26] investigated three CEFR levels (A2, B1, B2), Ehret and Szmrecsanyi [50] analyzed six proficiency levels categorized according to the amount of L2 instructional exposure, while the present study focused on four CEFR levels (A1, A2, B1, B2). A third plausible reason could stem from variation in the L2 English speakers' L1 backgrounds across the studies. Wang et al. [26] looked at only at Chinese L2 English learners, and Ehret and Szmrecsanyi [50] included L2 English learners from 10 different L1 backgrounds most of which were Indo-European languages. In contrast, the current study examined L2 English learners from ten L1 backgrounds including some non Indo-Euorepan languages (i.e., Arabic, Japanese, Mandarin, Turkish). Although these potential reasons could have led to the observed different results, more research is needed to account for the non-linear debelopmental patterns for the L2 English speakers in the current study.

Overall, findings from the analysis of Kolmogorov complexity across L2 proficiency levels within four distinct L2 speaker groups indicated that Kolmogorov metrics are cross-linguistically flexible as they can differentiate between proficiency levels within each examined L2 group. Further research is needed to explain the observed developmental trends captured by

Kolmogorov complexity metrics, preferably by comparing their results with traditional complexity metrics.

## Kolmogorov complexity across L1

Existing L2 writing complexity studies have examined the L1 effect mainly at the syntactic level [47, 48], e.g., [49] or across a few L1 backgrounds e.g., [50]. The current study is one of the few studies that investigated the L1 effect on L2 writing complexity at both the syntactic and morphological levels and across a wider range of L1 backgrounds. Results from the analysis of L2 English texts by ten L1 backgrounds confirmed earlier findings from fine-grained complexity metrics showing that L1-related variation is overlooked when L1 background is not considered in the analysis e.g., [48].

The analysis of L2 English texts across and between L1 groups made it possible to detect L1-based differences in syntactic and morphological complexity in written L2 production. Two main differences were observed in the syntactic complexity metric results. First, the aggregated results for L2 English writing suggested that L2 English learners generally produced more syntactic complex structures at level B1 than A2. In contrast, the L1-based analysis showed that Italian, Portuguese and Turkish L1 speakers did not significantly increase their use of complex syntactic structures as they moved from A2 to B1. Second, the combined findings indicated that B2 level English learners produced less syntactically complex texts than those at the B1 level. The L1-centered results revealed that Turkish L1 learners did not experience the same trend. Turkish L2 learners of English wrote more syntactically complex texts at the B2 level than the B1 level. Overall, although some L1 groups followed similar syntactic complexity patterns, this study found some L1-based differences in the use of complex syntactic sentences in L2 English writing. This corroborates previous findings using typical complexity measures [48], e.g., [49].

In terms of morphological complexity, three findings did not overlap in the combined versus separate analysis of L2 English essays. First, the aggregate analysis indicated a considerable decrease in the use of complex morphological features from A1 to A2. However, a less sharp decrease is observed in Arabic, French, Japanese, Mandarin, and Portuguese L2 learners' A2 essays relative to the A1 level. These learners did not substantially increase their use of inflected word forms when progressing from A1 to A2.

Second, the combined results showed a substantial increase in the production of morphologically complex essays at the upper-intermediate level (B2) than the lower intermediate level (B1). In contrast, L1-based analysis indicated that Italian L1 speakers showed a smaller significant increase at the B2 level, while Turkish L1 speakers did not show this increase in the same direction. Pairwise comparisons revealed that Turkish-English learners tended to produce significantly more morphologically complex writing at B1 than at B2 ($z = 4.08$, $p < 0.001$).

A third observed difference lies in the developmental patterns of morphological complexity. The combined analysis suggested a U-shaped development in the use of complex morphology, with A1 and B2 level English learners using more complex morphological forms than those at the A2 and B1CEFR levels. The L1-informed analysis showed that not all L1 groups followed the same morphological complexity trajectory. The L1 Turkish group demonstrated a continued decrease in the use of complex morphological features with increasing L2 English proficiency, suggesting their divergence from the observed U-shaped development for morphological complexity.

The observed L1-based differences are in line with the view that language development is variable across different learner groups [41]. In this account, language acquisition and processing are dynamic and variable, shaped by various factors including the learners' L1. These

L1-based variations can manifest in the use of specific syntactic structures and morphological features in L2 writing. In the present study, L2 English speakers from non-Indo-European L1 backgrounds (i.e., Arabic, Turkish) tended to use sentences with less rigid orders at the early proficiency stages (A1, A2). It is known that both of Arabic and Turkish have a relatively free word order, which might have led learners from these two languages to transfer their L1 word order rules to their L2 English writing [89, 90]. Meanwhile, L2 English speakers from Indo-European L1 backgrounds (i.e., Italian, Portuguese) showed a similar decrease in syntactic complexity (i.e., more sentences with free word orders) at the B1 CEFR level. Likewise, Italian and Portuguese are characterized by their flexible word order [91], which might have prompted their L1 speakers to employ non-canonical word order in their L2 English composition. Even though these findings suggest an L1 influence, they cannot pinpoint which specific linguistic features were distinct across the L1 backgrounds. Triangulating these results with traditional complexity metrics might reveal insightful findings.

Converging evidence from global complexity measures [47], fine-grained measures [48], and an information-theoretic measure support this L1-based variability in learner writing. However, much less attention has been paid to the effects of L1 on the realization of L2 complexity in writing. Current findings highlight the need to consider the role of L1 background when assessing complexity in L2 writing. This convergence also points out the validity of Kolmogorov complexity and positions it as a promising tool especially in the preliminary analysis of less researched languages, which could aid researchers in capturing general complexity trends in the data that would facilitate a more fine-grained analysis of these trends.

## Use of Kolmogorov complexity

Current findings indicated that two Kolmogorov complexity metrics, including syntactic and morphological complexity can significantly distinguish among CEFR proficiency levels within different language groups and across L1 backgrounds. Meanwhile, only numerical results suggested that overall Kolmogorov complexity could discriminate CEFR proficiency categories. The present results are consistent with the observation that complexity trajectories are non-linear e.g., [17, 45, 46], as indicated by the U-shaped pattern for morphological complexity in the aggregated L2 English data. Further, in line with Larsen-Freeman's [41] dynamic systems framework, the L1 effect findings suggested that writing complexity tends to follow distinct developmental trajectories for L2 learners from different L1 groups. Overall, the compression complexity measure seems capable of delineating proficiency stages within a range of languages and across L1 profiles. This study showed that an algorithmic, information-theoretic measure of language complexity can successfully be used to distinguish between essays written by L2 learners at different proficiency levels.

## Limitations and future directions

Although this study revealed the potential of a relatively new L2 complexity measure, it has a few limitations. First, this study examined complexity in four Indo-European languages which, despite their observed differences, share common features (e.g., the same writing script). This is because most of the CEFR rated learner corpora focus on L2 Indo-European languages [92], e.g., [93], and the presence of an objective measure of L2 proficiency was necessary to examine the L2 effect in a robust manner. Investigating the utility of this information-theoretic complexity measure for L2 production in other non-Indo-European languages such as Chinese, Korean, Japanese and Arabic remains worthwhile. Second, Kolmogorov complexity relies on a compression technique, and it remains unknown which language patterns and strings the compression programs (e.g., gzip) use to generate a compressed text version. This

algorithmic complexity measure might be difficult to interpret linguistically. Future research needs to address this topic to facilitate the linguistic interpretation of Kolmogorov complexity. Third, Kolmogorov complexity is a global measure and does not pinpoint the exact language structures that increase/reduce complexity scores. It might be desirable to conduct correlational analyses between Kolmogorov complexity and established measures [36, 54] to better understand Kolmogorov complexity results. Finally, the inclusion of traditional measures along with Kolmogorov complexity e.g., [26] could foster a better understanding of L1 based differences.

## Conclusion

This study examined the ability of Kolmogorov complexity metrics to distinguish among L2 proficiency levels within a number of languages (Czech, German, Italian, English) and across L1 backgrounds using two large and CEFR rated L2 corpora. Results indicated that Kolmogorov syntactic and morphological complexity metrics can discriminate CEFR levels within various L2s and L1 backgrounds. Interestingly, while Kolmogorov complexity revealed that some L2 groups (Czech, German, Italian) exhibited a somewhat a linear development for syntactic and morphological complexity, a non-linear pattern was found in the L2 English data. These findings could suggest that L2 language development is likely non-sequential, consistent with Larsen-Freeman's [41] dynamic systems framework. However, these findings are preliminary and complementary research using other established complexity metrics is needed to support the current observations.

This study could offer tentative implications for L2 writing research. The observed L1-based differences call for more scholarly attention to the potential role of L1 background on the development of syntactic and morphological complexity. Crucially, findings from studies using various complexity measures (global, fine-grained, information-theoretic) have supported the influence of L1 on L2 writing [47–50]. Thus, treating learners of the same L2 as a homogenous group risks overlooking important developmental patterns in their L2 written productions, potentially preventing us from gaining valuable insights into how linguistic complexity develops across L2 proficiency stages. While several studies using traditional measures have explored the L1 effect in the L2 syntactic complexity research [47, 48], e.g., [49], limited comparable interest has been found in the L2 morphological complexity research. Further investigation of the influence of L1 background on L2 writing complexity remains needed.

The main advantage of Kolmogorov complexity, a language-general and objective complexity index, is that it could facilitate the investigation of less researched languages in the L2 writing field. However, Kolmogorov complexity measures are not expected to replace existing syntactic measures, they are introduced here to provide another useful tool for examining L2 learners' syntactic development. The L2 writing field would benefit from the availability of a wider range of L2 complexity measures, each targeting language at a distinct level of granularity. Finally, although several studies examined the utility of Kolmogorov complexity measures in L2 writing production, whether it can similarly detect linguistic complexity in L2 spoken production data remains unknown.

## Author Contributions

**Conceptualization:** Alaa Alzahrani.

**Data curation:** Alaa Alzahrani.

**Formal analysis:** Alaa Alzahrani.

**Investigation:** Alaa Alzahrani.

**Methodology:** Alaa Alzahrani.

**Visualization:** Alaa Alzahrani.

**Writing – original draft:** Alaa Alzahrani.

**Writing – review & editing:** Alaa Alzahrani.

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
