## [Decision Letter · Decision Letter 0]

13 Feb 2024

PONE-D-24-00299Kolmogorov measures for L2 writing complexity: A cross-linguistic perspectivePLOS ONE

Dear Dr. Alzahrani,

Thank you for submitting your manuscript to PLOS ONE. After careful consideration, we feel that it has merit but does not fully meet PLOS ONE’s publication criteria as it currently stands. Therefore, we invite you to submit a revised version of the manuscript that addresses the points raised during the review process.

We look forward to receiving your revised manuscript.

Kind regards,

Lawrence Jun Zhang, Ph.D.

Academic Editor

PLOS ONE

Journal Requirements:

Reviewers' comments:

Reviewer's Responses to Questions

**Comments to the Author**

1. Is the manuscript technically sound, and do the data support the conclusions?

Reviewer #1: Yes

Reviewer #2: Yes

2. Has the statistical analysis been performed appropriately and rigorously? 

Reviewer #1: Yes

Reviewer #2: Yes

3. Have the authors made all data underlying the findings in their manuscript fully available?

Reviewer #1: Yes

Reviewer #2: Yes

4. Is the manuscript presented in an intelligible fashion and written in standard English?

Reviewer #1: Yes

Reviewer #2: Yes

5. Review Comments to the Author

Reviewer #1: General comment:

This study aims to investigate the differences in L2 syntactic complexity in the writings of learners with different proficiency levels and L1 backgrounds. The author contends that previous complexity measures are unreliable due to their susceptibility to the influence of learner errors and their “language-specific” nature. To address this, the author suggests using Kolmogorov complexity as a more robust and reliable measure of syntactic complexity. The study concludes that Kolmogorov complexity could be a valuable metric in L2 writing research due to its cross-linguistic flexibility and holistic nature. While the study is well-designed and its findings have important implications for L2 complexity research, there are some major issues in the manuscript that need to be addressed.

Specific comments:

1. The main issue in this manuscript is the critique of current syntactic complexity measures, such as the mean length of sentence and clause (MLS and MLC). Much of the criticism is not entirely accurate. For instance, the author asserts that "common complexity metrics in L2 writing research are either tailored to specific languages… (p. 31)." The author should be cautious when using the term "language-specific," as the majority of existing complexity measures are, in fact, language-neutral. Many measures (e.g., MLS and MLC) can be utilized to assess syntactic complexity in writing across various second languages (L2s), apart from L2 English. Furthermore, numerous studies have sought to investigate the applicability of these traditional measures in evaluating L2 writing development in L2 Chinese, French, Dutch and German, to name a few. As argued by the author, the main problem lies in the lack of tools to automatically calculate the scores of these measures for other L2s. Therefore, it is inappropriate to claim that the measures themselves are language-specific.

Similarly, while it is true that grammatical units such as clause and T-unit may be defined and operationalized differently across various L2s, it is inaccurate to regard the way these units are defined and operationalized as "subjective." Usually, these units are counted in an objective way according to a certain working definition of the target unit (e.g., Lu, 2011). In summary, the author should present the shortcomings of existing syntactic complexity measures more accurately.

2. As previously mentioned, a limitation of the existing studies is that they rely on those instruments that cannot be directly reused to analyze syntactic complexity in the writings of other L2s. Thus, it is assumed that this may not be the case for Kolmogorov complexity measures. It is suggested that there exists a tool that is capable of calculating the scores of Kolmogorov complexity measures for all L2s. Therefore, it is essential for the author to provide a detailed description of this tool or instrument in the manuscript.

3. p. 18 “Significance testing was performed for two complexity metrics: syntactic and morphological Complexity”

Is there a specific reason to discard the overall complexity metric for the significance test?

4. I agree that Kolmogorov measures have their own merits compared to traditional complexity measures. However, it does not necessarily mean that these measures are perfect or universally applicable to all research on L2 complexity. For instance, in this study, Kolmogorov measures are used to investigate L1 influence on writing performance. The author concludes that there are "L1-based differences in the use of complex syntactic sentences in L2 writing (p. 35)." However, this finding may be too general. It is important to identify specific differences among the writings by students from different L1 backgrounds and determine to what extent those differences may be caused by linguistic differences in the L1s. These questions should be the primary focus of studies on L1 influence. Traditional measures, especially fine-grained measures, may be more likely to answer these questions than Kolmogorov measures. I am eager to see the author's justifications for this comment.

Reviewer #2: This study examined the differences in Kolmogorov complexity between L2 proficiency levels across distinct languages and across different L1s. I think it is interesting and significant to consider the potential influence of L1 background on L2 writing complexity. However, I have some main concerns at both of the conceptual and methodological levels.

Regarding RQ1, which aims to compare the differences in Kolmogorov complexity across different L2 proficiency levels (A1 and B1) in four languages (German, Italian, Czech, and English), I find the rationale for this comparison unclear. It is evident that there would be cross-linguistic differences in Kolmogorov complexity due to variations in language structures and features among the four languages. However, the significance of this comparison in the context of L2 proficiency development remains ambiguous.

Additionally, I don’t see the rationale for discussing the cross-linguistics differences with the topics, settings, and writing genres uncontrolled. I suggest a clearer explanation of the rationale behind RQ1 and a more explicit description of the methodology to highlight the study’s significance.

Moving on to RQ2, which explores the impact of L1 background on L2 English writing Kolmogorov complexity, I question the choice of using Kolmogorov complexity metrics instead of traditional syntactic complexity metrics. Given that English is the target language for RQ2, it seems more appropriate to utilize the traditional commonly used syntactic complexity metrics for this analysis. Furthermore, the discussion of L1 effects on L2 writing complexity lacks depth and fails to fully explore the potential implications of the findings based on the diverse L1 backgrounds examined. The methodology lacks details regarding the selection criteria for written essays, including topics, settings, and writing genres, which are crucial factors that could influence the observed differences in Kolmogorov complexity. I recommend a thorough discussion of the potential L1 effects based on the diverse L1 backgrounds and a clearer description of the essays selection process from the corpus.

Additionally, I suggest updating the literature review to include recent studies on syntactic complexity validation that are relevant to the study's objectives. For example, Xu, T. S., Zhang, L. J., & Gaffney, J. S. (2023). A multidimensional approach to assessing the effects of task complexity on L2 students’ argumentative writing. Assessing Writing, 55, 100690, 1-16. https://doi.org/10.1016/j.asw.2022.100690 By incorporating these recent findings, the study can provide a more comprehensive review of the existing literature and strengthen its theoretical framework.

Finally, in the conclusion, there is a need for expansion, as there appears to be a lack of discussion regarding the significance and implications of the study’s findings.

6. PLOS authors have the option to publish the peer review history of their article (what does this mean?). If published, this will include your full peer review and any attached files.

Reviewer #1: **Yes: **Peng Bi

Reviewer #2: No

---

## [Author Response · Author response to Decision Letter 0]

21 Feb 2024

I would like the editor-in-chief and the reviewers for their time and much needed comments. I hope I have sufficiently addressed all the comments, and that the manuscript has somewhat improved. Please find below my reply (in black) to each reviewer’s comment (in blue).

Reviewer #1: 

Specific comments:

1. The main issue in this manuscript is the critique of current syntactic complexity measures, such as the mean length of sentence and clause (MLS and MLC). Much of the criticism is not entirely accurate. For instance, the author asserts that "common complexity metrics in L2 writing research are either tailored to specific languages… (p. 31)." The author should be cautious when using the term "language-specific," as the majority of existing complexity measures are, in fact, language-neutral. Many measures (e.g., MLS and MLC) can be utilized to assess syntactic complexity in writing across various second languages (L2s), apart from L2 English. Furthermore, numerous studies have sought to investigate the applicability of these traditional measures in evaluating L2 writing development in L2 Chinese, French, Dutch and German, to name a few. As argued by the author, the main problem lies in the lack of tools to automatically calculate the scores of these measures for other L2s. Therefore, it is inappropriate to claim that the measures themselves are language-specific.

I thank the reviewer for sharing this insightful comment. I agree with the reviewer in this regard, and as was suggested, the measures themselves are not language-specific but rather the avaliable automatic tools are. To clarify this point, I revised the relevant parts in the manuscript.

Similarly, while it is true that grammatical units such as clause and T-unit may be defined and operationalized differently across various L2s, it is inaccurate to regard the way these units are defined and operationalized as "subjective." Usually, these units are counted in an objective way according to a certain working definition of the target unit (e.g., Lu, 2011). In summary, the author should present the shortcomings of existing syntactic complexity measures more accurately.

I agree with the reviewer. As was suggested, T-units are counted in an objective way according to a certain definition. Thus, it would be better to state that these units are objective but may sometimes not fit learners’ productions. The manuscript was revised in line with this idea.

2. As previously mentioned, a limitation of the existing studies is that they rely on those instruments that cannot be directly reused to analyze syntactic complexity in the writings of other L2s. Thus, it is assumed that this may not be the case for Kolmogorov complexity measures. It is suggested that there exists a tool that is capable of calculating the scores of Kolmogorov complexity measures for all L2s. Therefore, it is essential for the author to provide a detailed description of this tool or instrument in the manuscript.

Thank you for sharing this suggestion. I included the following subsection to address this comment:

“Kolmogorov complexity calculation tool

To our knwoledge, there is currently no specialized software that automatically calculates Kolmogorov complexity metrics. These metrics were calculated in this study using R scripts developed and shared publicly by Ehret (2016). This calculation is done in several steps. First, the free compression program gzip must be downloaded and installed. Second, the corpus files should be stored in txt format. Third, Ehret’s R scripts should be run, which would automatically calculate Kolmogrov complexity metrics. Further details about these R scripts can be found here: https://github.com/katehret/measuring-language-complexity.”

3. p. 18 “Significance testing was performed for two complexity metrics: syntactic and morphological Complexity” Is there a specific reason to discard the overall complexity metric for the significance test?

To answer the reviewer’s question, yes there was a specific reason to discard this metric for significance testing. Specifically, I attempted to enter overall complexity metric into a statistical test, but the results were unreliable. Overall complexity metric is calculated using regression residuals, which provide one value per group (e.g., level A2 group, level B1 group...etc). Running a statistical test to compare between two values is typically not recommended as it would result in incorrect estimates. Thus, only raw scores were reported in the present study. This reasoning was included in the manuscript.

4. I agree that Kolmogorov measures have their own merits compared to traditional complexity measures. However, it does not necessarily mean that these measures are perfect or universally applicable to all research on L2 complexity. For instance, in this study, Kolmogorov measures are used to investigate L1 influence on writing performance. The author concludes that there are "L1-based differences in the use of complex syntactic sentences in L2 writing (p. 35)." However, this finding may be too general. It is important to identify specific differences among the writings by students from different L1 backgrounds and determine to what extent those differences may be caused by linguistic differences in the L1s. These questions should be the primary focus of studies on L1 influence. Traditional measures, especially fine-grained measures, may be more likely to answer these questions than Kolmogorov measures. I am eager to see the author's justifications for this comment.

I agree with the reviewer, specific syntactic differences in L2 writings are more important for linguists than general differences. I would like to argue that the use of Kolmogorov complexity might be the initial step in the linguistic analysis especially when working with less researched languages. The preliminary Kolmogorov results would pinpoint the presence of overall syntactic/morphological differences between L2 proficiency levels within the same language. In the second step, traditional fine-grained measures might be used to investigate in more detail these syntactic differences. The use of several syntactic measures has the potential of providing converging evidence for the observed syntacic complexity development. As such, Kolmogorov complexity measures are not expected to replace existing syntactic measures, rather they are introduced here to provide yet another useful tool in examining L2 learners’ syntactic development. The L2 writing field would benefit from the availability of a wider range of L2 complexity measures, each targeting language at a distinct level of granularity.

To address the reviewer’s comment, the limitation section highlighted that Kolmogorov metrics do not pinpoint the specific differences, but rather generally indicate that differences exist. Thus, more work is needed to compare between Kolmogorov complexity results and traditional metrics.

Reviewer #2: 

1. Regarding RQ1, which aims to compare the differences in Kolmogorov complexity across different L2 proficiency levels (A1 and B1) in four languages (German, Italian, Czech, and English), I find the rationale for this comparison unclear. It is evident that there would be cross-linguistic differences in Kolmogorov complexity due to variations in language structures and features among the four languages. However, the significance of this comparison in the context of L2 proficiency development remains ambiguous.

Thank you for sharing this comment. I would like to clarify that the rationale for this comparison was to assess the applicability of Kolmogorov complexity as a tool for measuring linguistic complexity, regardless of specific language characteristics. 

To address this comment, I added the following rationale for the research questions in the manuscript:

“As it is argued that the syntactic complexity of L2 writings develops with increasing L2 proficiency (e.g., Khushik & Huhta, 2020; Martínez, 2018; Ouyang et al., 2022; Wang et al., 2022), this study investigated the reliability of Kolmogorov complexity by investigating whether it can capture the expected syntactic differences across proficiency levels within various L2s (RQ1) and across L1 backgrounds (RQ2). The rationale for RQ1 is to establish the cross-linguistic flexibility of Kolmogorov complexity by examining whether it can capture syntactic development across L2 proficiency stages in a wider set of languages rather than a single language. The motivation for RQ2 is to assess whether Kolmogorov complexity can detect L1-based differences in L2 writings similar to non-information theoretic complexity measures (e.g., S. A. Crossley & McNamara, 2012; Lu & Ai, 2015; Phuoc & Barrot, 2022).”

I agree with the reviewer that differences in Kolmogorov complexity could be attributed to variations in language structures in the four examined languages. To clarify this point in the manuscript, the discussion section was revised so as to not draw cross linguistic comparisons between the results of the different L2s.

As for conducting this comparison in the context of L2 proficiency, it is expected that L2 writings produced by learners at different L2 proficiency levels would show variation in syntactic complexity (e.g., Khushik & Huhta, 2020; Martínez, 2018; Ouyang et al., 2022; Wang et al., 2022). This study investigated whether Kolmogorov complexity can capture the expected syntactic differences across proficiency levels. This would examine the utility of Kolmogorov complexity metrics and provide one advantage for their use in the L2 writing literature.

2. Additionally, I don’t see the rationale for discussing the cross-linguistics differences with the topics, settings, and writing genres uncontrolled. I suggest a clearer explanation of the rationale behind RQ1 and a more explicit description of the methodology to highlight the study’s significance.

I agree with the reviewer. Now the cross-linguistic comparisons are removed from the manuscript.

The rationale for RQ1 is included in the revised version. A more detailed description of the corpora is also provided in the revised manuscript.

3. Moving on to RQ2, which explores the impact of L1 background on L2 English writing Kolmogorov complexity, I question the choice of using Kolmogorov complexity metrics instead of traditional syntactic complexity metrics. Given that English is the target language for RQ2, it seems more appropriate to utilize the traditional commonly used syntactic complexity metrics for this analysis. 

I agree with the reviewer, traditional complexity metrics are also valid for examining RQ2 and can be even more appropraite than Kolmogrov complexity for this type of analysis. However, as the current study aimed to investigate the effectiveness of Kolmogrov complexity, the analysis for RQ2 was primarily conducted using Kolmogrov complexity. Nevertheless, I believe that a more comprehensive analysis of L1-effects might require the use of both traditional and Kolmogrov measures to account for L1-based differences. This last point is now included in the limitation section.

4. Furthermore, the discussion of L1 effects on L2 writing complexity lacks depth and fails to fully explore the potential implications of the findings based on the diverse L1 backgrounds examined. 

As suggested by the reviewer, now the discussion section includes a new paragraph elaborating on the L1 effects on L2 writing complexity. This was added to the discussion section:

“The observed L1-based differences are in line with the view that language development is variable across different learner groups (Larsen-Freeman, 2006). In this account, language acquisition and processing are dynamic and variable, shaped by various factors including the learners’ L1. These L1-based variations can manifest in the use of specific syntactic structures and morphological features in L2 writing. In the present study, L2 English speakers from non-Indo-European L1 backgrounds (i.e., Arabic, Turkish) tended to use sentences with less rigid orders at the early proficiency stages (A1, A2). It is known that both of Arabic and Turkish have a relatively free word order, which might have led learners from these two languages to transfer their L1 word order rules to their L2 English writing (Kornfilt, 2003; Ryding, 2014). Meanwhile, L2 English speakers from Indo-European L1 backgrounds (i.e., Italian, Portuguese) showed a similar decrease in syntactic complexity (i.e., more sentences with free word orders) at the B1 CEFR level. Likewise, Italian and Portuguese are characterized by their flexible word order (Arnaiz, 1998), which might have prompted their L1 speakers employ non-canonical word order in their L2 English composition. Even though these findings suggest an L1 influence, they cannot pinpoint which specific linguistic features were distinct across the L1 backgrounds. Triangulating these results with traditional complexity metrics might reveal insightful findings.”

5. The methodology lacks details regarding the selection criteria for written essays, including topics, settings, and writing genres, which are crucial factors that could influence the observed differences in Kolmogorov complexity. I recommend a thorough discussion of the potential L1 effects based on the diverse L1 backgrounds and a clearer description of the essays selection process from the corpus.

I thank the reviewer for sharing this important observation. As for the discussion of potential L1 effects, a paragraph was added to the discussion section examining the potential role of L1 background on L2 writing complexity. 

Additionally, a clearer description of the essay selection process from the corpora is discussed in the revised manuscript. This paragraph was added:

“Written essays were selected based on two factors: L2 proficiency, and genre. For MERLIN, only texts rated A1 and B1 were selected as the other CEFR levels were not equally represented across L2. These texts fall in the descriptive and expository genre. For EFCAMDAT, as this corpus comprises a large number of texts across four CEFR levels (A1, A2, B1, B2), they were all included. The selected essays from EFCAMDAT were all persuasive essays. Since EFCAMDAT is semi-longitudinal (Shatz, 2020), texts written by the same L2 learner were removed to reduce idiosyncratic complexity features. While differences across corpora exist, this study focused on analyzing L2 writing complexity patterns within each L2 group.”

6. Additionally, I suggest updating the literature review to include recent studies on syntactic complexity validation that are relevant to the study's objectives. For example, Xu, T. S., Zhang, L. J., & Gaffney, J. S. (2023). A multidimensional approach to assessing the effects of task complexity on L2 students’ argumentative writing. Assessing Writing, 55, 100690, 1-16. https://doi.org/10.1016/j.asw.2022.100690 By incorporating these recent findings, the study can provide a more comprehensive review of the existing literature and strengthen its theoretical framework.

As was suggested, the literature review is updated and now includes the recent studies on syntactic complexity validation. The suggested reference is now included in the paper.

7. Finally, in the conclusion, there is a need for expansion, as there appears to be a lack of discussion regarding the significance and implications of the study’s findings.

I thank the reviewer for sharing this point. As suggested, now the conclusion section is expanded to include a discussion of the significance and implications of the study’s findings. This part was included in the conclusion to address this comment:

“This study could offer tentative implications for L2 writing research. The observed L1-based differences call for more scholarly attention to the potential role of L1 background on the development of syntactic and morphological complexity. Crucially, findings from studies using various complexity measures (global, fine-grained, information-theoretic) have supported the influence of L1 on L2 writing (S. A. Crossley & McNamara, 2012; Ehret & Szmrecsanyi, 2019; Lu & Ai, 2015; Phuoc & Barrot, 2022). Thus, treating learners of the same L2 as a homogenous group has the risk overlooking important developmental patterns in their L2 written productions, potentially preventing us from gaining valuable insights into how linguistic complexity develops across L2 proficiency stages. While several studies using traditional measures have explored the L1 effect in the L2 syntactic complexity research (e.g., S. A. Crossley & McNamara, 2012; Lu & Ai, 2015; Phuoc & Barrot, 2022), limited comparable interest has been found in the L2 morphological complexity research. More examination of the role of L1 in L2 writing complexity is needed.

The main advantage of Kolmogorov complexity, a language-general and objective complexity index, is that it could facilitate the investigation of less researched languages in the L2 writing field. However, Kolmogorov complexity measures are not expected to replace existing syntactic measures, rather they are introduced here to provide yet another useful tool in examining L2 learners’ syntactic development. The L2 writing field would benefit from the availability of a wider range of L2 complexity measures, each targeting language at a distinct level of granularity. “

---

## [Decision Letter · Decision Letter 1]

17 Mar 2024

PONE-D-24-00299R1Utility of Kolmogorov complexity measures: analysis of L2 groups and L1 backgroundsPLOS ONE

Dear Dr. Alzahrani,

Thank you for submitting your manuscript to PLOS ONE. After careful consideration, we feel that it has merit but does not fully meet PLOS ONE’s publication criteria as it currently stands. Therefore, we invite you to submit a revised version of the manuscript that addresses the points raised during the review process.

We look forward to receiving your revised manuscript.

Kind regards,

Lawrence Jun Zhang, Ph.D.

Academic Editor

PLOS ONE

Journal Requirements:

Reviewers' comments:

Reviewer's Responses to Questions

**Comments to the Author**

1. If the authors have adequately addressed your comments raised in a previous round of review and you feel that this manuscript is now acceptable for publication, you may indicate that here to bypass the “Comments to the Author” section, enter your conflict of interest statement in the “Confidential to Editor” section, and submit your "Accept" recommendation.

Reviewer #1: All comments have been addressed

Reviewer #2: All comments have been addressed

2. Is the manuscript technically sound, and do the data support the conclusions?

Reviewer #1: Yes

Reviewer #2: Partly

3. Has the statistical analysis been performed appropriately and rigorously? 

Reviewer #1: Yes

Reviewer #2: Yes

4. Have the authors made all data underlying the findings in their manuscript fully available?

Reviewer #1: Yes

Reviewer #2: Yes

5. Is the manuscript presented in an intelligible fashion and written in standard English?

Reviewer #1: Yes

Reviewer #2: Yes

6. Review Comments to the Author

Reviewer #1: (No Response)

Reviewer #2: Most of my comments have been addressed by the author. However, some minor issues still need to be considered.

1. Regarding the Method-corpus section, the author introduced the corpus as "...,whereas EFCAMDAT also involved persuasive texts (e.g., selling items in an online auction)." The statement is unclarified. I suggest further explain whether EFCAMDA exclusively comprised persuasive genre texts or if the author specifically selected persuasive texts from a diverse range of genres.

2. In the third paragraph of the Method-corpus section, the phrase "For MERLIN, only texts rated A1 and B1 were selected as the other CEFR levels were not equally represented across L2" should be revised for better understanding.

3. Also in the third paragraph of the Method-corpus part, the author mentioned “Since EFCAMDAT is semi-longitudinal (Shatz, 2020), texts written by the same L2 learner were removed to reduce idiosyncratic complexity features.” Could you further explain the removal of texts written by the same L2 learner? How to remove? To remove identical essays by individual writers or all essays produced by a single writer?

4. I still think it is necessary to include the traditional measures to account for L1 based differences (RQ2). However, I did not locate this point in the limitation section, which I think should be explicitly addressed in the manuscript.

5.Please carefully proofread the manuscript.

7. PLOS authors have the option to publish the peer review history of their article (what does this mean?). If published, this will include your full peer review and any attached files.

Reviewer #1: No

Reviewer #2: No

---

## [Author Response · Author response to Decision Letter 1]

21 Mar 2024

I thank the reviewers for their very helpful comments. In the following, I included each comment followed by my reply.

1. Regarding the Method-corpus section, the author introduced the corpus as "...,whereas EFCAMDAT also involved persuasive texts (e.g., selling items in an online auction)." The statement is unclarified. I suggest further explain whether EFCAMDA exclusively comprised persuasive genre texts or if the author specifically selected persuasive texts from a diverse range of genres.

Thank you for pointing out this point. I clarified this part as follows:

“…, whereas EFCAMDAT largely comprised persuasive texts (e.g., selling items in an online auction). These were the main genres in each corpus.”

2. In the third paragraph of the Method-corpus section, the phrase "For MERLIN, only texts rated A1 and B1 were selected as the other CEFR levels were not equally represented across L2" should be revised for better understanding.

I modified the sentence as follows:

“For MERLIN, this study selected texts with A1 and B1 ratings because texts with higher or lower CEFR ratings were not equally represented across L2 in this corpus.”

I hope it’s easier to read and understand now.

3. Also in the third paragraph of the Method-corpus part, the author mentioned “Since EFCAMDAT is semi-longitudinal (Shatz, 2020), texts written by the same L2 learner were removed to reduce idiosyncratic complexity features.” Could you further explain the removal of texts written by the same L2 learner? How to remove? To remove identical essays by individual writers or all essays produced by a single writer?

Thank you for sharing this suggestion. I revised this part as follows:

“EFCAMDAT is semi-longitudinal and contains several texts written by the same L2 learner (Shatz, 2020). This study selected only one text per L2 learner to reduce idiosyncratic complexity features.”

4. I still think it is necessary to include the traditional measures to account for L1 based differences (RQ2). However, I did not locate this point in the limitation section, which I think should be explicitly addressed in the manuscript.

I agree with the reviewer. This point is now mentioned in the limitation section as follows:

“Finally, the inclusion of traditional measures along with Kolmogorov complexity (e.g., Wang et al., 2022) could foster a better understanding of L1 based differences.”

5. Please carefully proofread the manuscript.

Thank you for your suggestions. The manuscript has been proofread for grammar and word choice. I hope it has much improved now.

---

## [Editor Report · Decision Letter 2]

24 Mar 2024

Utility of Kolmogorov complexity measures: analysis of L2 groups and L1 backgrounds

PONE-D-24-00299R2

Dear Dr. Alzahrani,

We’re pleased to inform you that your manuscript has been judged scientifically suitable for publication and will be formally accepted for publication once it meets all outstanding technical requirements.

Kind regards,

Lawrence Jun Zhang, Ph.D.

Academic Editor

PLOS ONE

---

## [Editor Report · Acceptance letter]

28 Mar 2024

PONE-D-24-00299R2 

PLOS ONE

Dear Dr. Alzahrani, 

I'm pleased to inform you that your manuscript has been deemed suitable for publication in PLOS ONE. Congratulations! Your manuscript is now being handed over to our production team.

Kind regards, 

on behalf of

Professor Lawrence Jun Zhang 

Academic Editor

PLOS ONE